# Next Generation of Cancer Drug Repurposing: Therapeutic Combination of Aspirin and Oseltamivir Phosphate Potentiates Gemcitabine to Disable Key Survival Pathways Critical for Pancreatic Cancer Progression

**DOI:** 10.3390/cancers14061374

**Published:** 2022-03-08

**Authors:** Bessi Qorri, Reza Bayat Mokhtari, William W. Harless, Myron R. Szewczuk

**Affiliations:** 1Department of Biomedical and Molecular Sciences, Queen’s University, Kingston, ON K7L 3N6, Canada; bessi.qorri@queensu.ca (B.Q.); reza.bayatmokhtari@sickkids.ca (R.B.M.); 2ENCYT Technologies Inc., Membertou, NS B1S 0H1, Canada

**Keywords:** repurposed drugs, multimodal therapy, cancer therapy, drug repositioning

## Abstract

**Simple Summary:**

Drug repurposing in combination with clinical standard chemotherapeutics opens a novel and promising clinical treatment approach for patients with pancreatic cancer. This report presents a novel therapeutic effect of the combination of aspirin and oseltamivir phosphate with chemotherapeutic gemcitabine as a treatment option for pancreatic cancer. The data suggest that targeting mammalian neuraminidase-1 on pancreatic cancer cells with these repurposed drugs is crucial for modulating cell proliferation, invasion, clonogenicity, and migration. These promising results warrant additional investigation to assess the potential of translating into the clinical setting to improve the cancer patient prognosis for an otherwise fatal disease.

**Abstract:**

Resistance to chemotherapeutics and high metastatic rates contribute to the abysmal survival rate in patients with pancreatic cancer. An alternate approach for treating human pancreatic cancer involves repurposing the anti-inflammatory drug, aspirin (ASA), with oseltamivir phosphate (OP) in combination with the standard chemotherapeutic agent, gemcitabine (GEM). The question is whether treatment with ASA and OP can sensitize cancer cells to the cytotoxicity induced by GEM and limit the development of chemoresistance. To assess the key survival pathways critical for pancreatic cancer progression, we used the AlamarBlue cytotoxicity assay to determine the cell viability and combination index for the drug combinations, flow cytometric analysis of annexin V apoptosis assay to detect apoptotic and necrotic cells, fluorometric QCM™ chemotaxis migration assay to assess cellular migration, fluorometric extracellular matrix (ECM) cell adhesion array kit to assess the expression of the ECM proteins, scratch wound assay using the 96-well WoundMaker™, and the methylcellulose clonogenic assay to assess clonogenic potential. The combination of ASA and OP with GEM significantly upended MiaPaCa-2 and PANC-1 pancreatic cancer cell viability, clonogenic potential, expression of critical extracellular matrix proteins, migration, and promoted apoptosis. ASA in combination with OP significantly improves the effectiveness of GEM in the treatment of pancreatic cancer and disables key survival pathways critical to disease progression.

## 1. Introduction

Pancreatic ductal adenocarcinoma (PDAC) accounts for the majority of pancreatic cancers [1]. Following diagnosis, there is an abysmal patient 5-year survival rate of only 7% [2,3]. Over the past five decades, there have been minimal improvements in treatment outcomes due to ineffective screening and early detection methods capable of identifying pancreatic cancer in a pre-malignant stage [4]. Clinical treatment options include surgical resection, neoadjuvant and adjuvant chemotherapy, radiation, and immunotherapy [5,6,7]. Unfortunately, heterogeneous cancer cell populations in primary tumors and secondary micro-metastases render them resistant to cytotoxic therapies [8,9]. Even if pancreatic cancer initially responds to chemotherapies such as gemcitabine (GEM), drug resistance inevitably develops in patients. One plausible mechanism may be from the tissue-damaging effects of chemotherapy, triggering the release of tissue repair molecules and the induction of an epithelial–mesenchymal transition (EMT) in the surviving cancer cell population fostering the enrichment of a cancer stem cell (CSC) population [10,11]. This theoretical proposed mechanism for the development of drug resistance and progression during chemotherapy treatment has been empirically demonstrated in several malignancies, including bladder and ovarian cancers, which may be amenable to therapeutic intervention [12,13]. Future therapies must target and disable the multiple biological mechanisms that drive PDAC progression and metastasis to overcome these pancreatic cancer treatment limitations. These cancer survival mechanisms include inflammatory and immune-derived promoters of tumor development and growth, acquired drug resistance mechanisms, and pro-metastatic signals in the tumor microenvironment (TME) that potentiate cancer cell dissemination and homing to distant organs [14].

Key cancer hallmarks of malignancy are not regulated by a single signaling pathway [14,15,16]. Mono- or multi-hallmark-targeting drugs have therapeutic advantages in targeting several pharmacological pathways and partially avoiding the development of drug resistance [14,17,18,19]. A recent review by Zhang et al. [20] describes the significant hurdles for discovering new drugs for cancer therapy in detail. They propose the necessitated development of an alternative strategy of drug repurposing, using old drugs for new therapeutic purposes. For example, Zhang and colleagues [21] investigated the anti-inflammatory drug aspirin (ASA) on PDAC cell lines. They found that ASA (a) increases the therapeutic efficacy of GEM by overcoming GEM-resistance and altering the expression of reprogramming factors, (b) inhibits the potential for self-renewal and enhances gemcitabine efficacy, (c) inhibits spheroid formation and aldehyde dehydrogenase isoform 1 (ALDH1) activity, which is defined as a marker for self-renewal capacity, (d) inhibits the development of primary CSC spheroids, (e) inhibits tumor growth and invasion in vivo, and (f) reduces the deposition of extracellular matrix (ECM) components, such as fibronectin and collagen. 

We recently reported on the missing link connecting the anti-cancer efficacy of ASA to the role of glycosylation in inflammation and tumorigenesis [22]. We showed that ASA exerts anti-cancer effects by targeting and inhibiting mammalian neuraminidase-1 (Neu-1). Neu-1 regulates the activation of several receptor tyrosine kinases (RTKs) [23] and TOLL-like receptors [24], and their downstream signaling pathways [25]. Neu-1 forms a complex with matrix metalloproteinase-9 (MMP-9) and G protein-coupled receptors (GPCRs), which are tethered to RTKs at the ectodomain [23]. Furthermore, oseltamivir phosphate (OP) has been reported to inhibit mammalian Neu-1 in complex with MMP-9 and GPCR tethered to several RTKs, many of which are overexpressed on cancer cells [26,27]. OP also downregulates several epidermal growth factor receptor (EGFR)-mediated pathways, such as the JAK/STAT, PI3K/Akt, and MAPK pathways, involved in cancer cell proliferation, metastasis, and tumor vascularization [28]. Previously, we have reported on the efficacy of OP monotherapy in mouse models of human ovarian [29], breast [30], and pancreatic cancers [23,28], and the mechanism of action of OP in regulating multistage tumorigenesis [25]. Using a mouse model of human MDA-MB-231 triple-negative breast cancer (TNBC) tumors, Haxho et al. [30] showed that OP treatment alone at 30 mg/kg daily, intraperitoneally reduced tumor vascularization and growth rate as well as significantly reduced tumor weight and metastatic migration to the lungs compared with the untreated cohorts. OP treatment at 50 mg/kg completely ablated tumor vascularization and growth and metastases to the lungs, with a significant survival rate at day 180 post-implantation, with complete tumor shrinking, and no relapses after 56 days off-drug.

Recently, Sambi et al. [31] investigated an alternative drug repurposing strategy, using ASA, metformin, and OP as a multimodal approach to control breast tumor growth and strategically prevent metastatic burden. They found that the triple combination of ASA, metformin, and OP administered with tamoxifen (Tmx) reduced cell proliferation, disabled Tmx chemoresistance, and increased apoptotic activity in monolayer and spheroid cultures of MDA-MB-231 TNBC cells and their Tmx-resistant variant. This triple combination of these repurposed drugs downregulated the acquisition of CSC-like properties of MDA-MB-231 cells and disrupted vasculogenic endothelial cell tube formation of human umbilical vein endothelial cells (HuVECs). For the first time, these findings demonstrated that the triple combination of ASA, metformin, and OP provided a practical multimodal therapeutic approach in targeting multistage tumorigenesis in TNBC.

In this present study, the combination of ASA and OP administered with GEM significantly upended MiaPaCa-2 and PANC-1 pancreatic cancer cell survival mechanisms, including viability, combination indices, clonogenic potential, the expression of critical extracellular matrix proteins, migration potential, and promoted apoptosis. Here, the repurposing of ASA and OP with GEM presents a strategic multi-therapeutic approach that disables multiple survival mechanisms of pancreatic cancer that promote progression and metastasis. 

## 2. Materials and Methods

### 2.1. Cell Lines

MiaPaCa-2 and PANC-1 pancreatic cancer cell lines were incubated in a standard cell incubator at 37 °C with 5% CO_2_. PANC-1 (ATCC^®^ CRL-1469™) is a human pancreatic cancer cell line, for which site of origin was the duct in a 56-year-old male with pancreatic ductal adenocarcinoma. PANC-1 has a genetic profile that has been characterized to express KRAS, TP53, and CDKN2A [28]. Mia-PaCa-2 cells (ATCC^®^ Number: CRL-1420™) are human pancreatic cancer cell lines with attached epithelial and floating rounded cells expressing the 17 beta-estradiol (E2)-binding estrogen receptor and derived from a male patient with carcinoma. HEK 293 cells are an immortalized cell line generated by the transfection of cultures of normal human embryonic kidney cells with sheared adenovirus 5 DNA. This hypotriploid cell line expresses an unusual cell surface receptor for vitronectin or the S-protein of the complement system, which is produced predominantly by the liver. Cells were maintained in DMEM supplemented with 10% fetal bovine serum (FBS), and 0.1% plasmocin. Cell lines were maintained in T-75 cell culture flasks up to 70–80% confluence prior to experimental use.

### 2.2. Reagents

Acetylsalicylic acid (>99% pure, Sigma–Aldrich, Steinheim, Germany) was dissolved in dimethyl sulfoxide (DMSO) to prepare a 550 mM stock solution, which was stored in aliquots at −20 °C. The highest used concentration of aspirin contains less than 0.5% *v*/*v* of DMSO in 1× PBS at a pH of 7. Oseltamivir phosphate-USP (batch No. MBAS20014A, >99% pure powder, Solara Active Pharma Sciences Ltd., New Mangalore-575011, Karnataka, India) was freshly dissolved in sterile normal saline before use, Gemcitabine hydrochloride (Sigma–Aldrich Canada Ltd.) was dissolved in PBS to create a 133.5 mM gemcitabine stock. This stock was serially diluted to produce 0.01 μM gemcitabine in 1× DMEM containing 10% fetal calf serum and 5 μg/mL plasmocin solution that was added to tissue culture flasks.

### 2.3. AlamarBlue Cytotoxicity Assay and Metabolic Activity

Drug-induced cytotoxicity was determined using the AlamarBlue assay, as previously described [32]. The MiaPaCa-2 and PANC-1 cells were seeded at 20,000 cells per well in flat-bottom 96-well plates. Cells were incubated overnight before being treated with ASA, OP, GEM, ASA+OP, ASA+GEM, OP+GEM, ASA+OP+GEM, or treated with DMSO as the untreated vehicle control. Cells were incubated at 37 °C at 5%CO_2_ for 24, 48, 72 h, or seven days for long-term treatment. Cytotoxicity was determined using the AlamarBlue reagent. 10 µL of AlamarBlue was added to every 100 µL of supernatant and incubated for 4 h at 37 °C and 5% CO_2_. The absorbance was recorded using (ex 560 nm; em 590 nm). Results were calculated by subtracting fluorescence from the blank media control and compared to the day 0 untreated control.

### 2.4. Combination Index

Combination index (CI) analyses were used to evaluate ASA/OP/GEM drug interactions in combination on MiaPaCa-2, PANC-1, and HEK 293 cells to classify interactions as synergistic, additive, or antagonistic. The resulting combination index (CI) theorem of Chou–Talalay offers quantitative definition for additive effect (CI = 1), synergism (CI < 1), and antagonism (CI > 1) in drug combinations [33]. Here, we used the formula of the sum of the ratio of the dose of each drug in the compound to the dose when used alone when the combination and compound produce 50%, 75%, and 95% efficacies. Using the following formula:$$\mathrm{CI}=\frac{D1}{\mathrm{Dx}1}+\frac{D2}{\mathrm{Dx}2}$$
where Dx1 and Dx2 represent concentrations of each drug alone to exert x% inhibitory effect and D1 and D2 are concentrations of the drugs in combination to elicit the same effect. A CI < 1 represents synergy, CI = 1 represents additive, and CI > 1 represents antagonism.

### 2.5. Flow Cytometric Analysis of Annexin V Apoptosis Assay

The Annexin V-FITC Apoptosis Detection Kit (BioVision; No. K101-25) was used to detect apoptotic and necrotic cells following treatment. Cells were pre-treated with ASA, OP, GEM, or their combination at the indicated concentration for seven days. Treated cells were collected and resuspended as 1 × 10^5^ cells in 500 μL of binding buffer. 5 μL of Annexin V-FITC and 5 μL of propidium iodide (PI) were added and incubated for 5 min at room temperature in the dark. Annexin V-FITC binding was analyzed by flow cytometry (Ex 488 nm; em 530 nm).

### 2.6. Migration Assay

The Fluorometric QCM™ Chemotaxis Migration Assay Kit (ECM510; Sigma-Aldrich Canada Co., Oakville, ON, Canada) was used to assess PANC-1 and MiaPaCa-2 cells migration following treatment with ASA, OP, and Gem, or a combination of the agents. Cells were treated for seven days in medium supplemented with 10% FBS for seven days every 48 h. Standard tissue culture media supplemented with FBS were added to the feeder tray. At the end of the 7th day of treatment, cells were trypsinized and resuspended in 1 x DMEM (no FBS) and added into the migration chamber and allowed to adhere. Cells were incubated for 4 h to allow migration in the media with FBS (acting as a chemotaxis agent). Unbound cells were removed, and the migration insert plate was transferred onto a new cell culture tray containing cell detachment solution. Cells were incubated at 37 °C for 30 min. The CyQuant GR dye was diluted in a 1:75 ratio of the 4× lysis buffer. It was added to the wells of the feeder tray containing the cell detachment solution with the cells that migrated through the membrane and incubated at room temperature for 15 min. The mixture was transferred to a new 96-well plate and was read with a fluorescence plate reader (Ex 450 nm; em 530 nm). 

### 2.7. Adhesion Assay

The Fluorometric Extracellular Matrix (ECM) Cell Adhesion Array Kit (Millipore, Darmstadt, Germany; ECM545) was used to assess the expression of the ECM proteins (collagen I, collagen II, collagen IV, fibronectin, laminin, tenascin, and vitronectin). The ECM Cell Adhesion Array kit utilizes a homogenous fluorescence detection format, allowing large-scale screening and quantitative comparison of multiple samples. Each kit contains a 96-well microtiter plate consisting of 12 × 8-well removable strips. Each well within a strip (7 wells in total) is pre-coated with a different ECM protein, along with one BSA-coated well (negative control). MiaPaCa-2 and PANC-1 cells were treated with single agents ASA, OP, and GEM, or a combination of the agents in a medium supplemented with 10% FBS for seven days every 48 h. To assess the expression of adhesion marker expression following drug treatment at the end of the 7th day, cells were trypsinized and resuspended (1 × 10^6^ cells/mL) and allowed to adhere onto coated substrates to capture adherent cells. Unbound cells were washed, and the adherent cells were lysed. CyQuant GR^®^ dye was diluted in a 1:75 ratio of 4× cell lysis buffer. 50 μL of the lysis buffer/dye solution was added to each well, and the plate was incubated at room temperature for 15 min with a cell lysis solution. The solution was mixed by pipetting, and 150 μL of the mixture was transferred to a 96-well plate read with a fluorescence plate reader (Ex 450 nm; em 530 nm).

### 2.8. Scratch Wound Assay

MiaPaCa-2 and PANC-1 pancreatic cancer cells were seeded in a flat bottom ImageLock™ 96-well plate at a density of 2 × 10^5^ cells/well in 100 µL culture medium and adhered overnight in an incubator at 37 °C and 5% CO_2_. The 96-well WoundMaker™ was used to create wounds in all wells simultaneously. Non-adherent cells were removed with a medium wash, and fresh media containing ASA, OP, GEM, or their combination at the indicated concentrations were added to the culture. The plate was placed in the IncuCyte ZOOM^®^, and the IncuCyte ZOOM^®^ software was used to schedule repeat scanning for every 2 h for 72 h, with the first scan to begin immediately. The assay plates were set to one image per well (Wide mode) and Scratch Wound scan type. The wound migration area was quantified using the IncuCyte ZOOM^®^ software comparing wound density at the indicated time point to the immediate wound density.

### 2.9. Methylcellulose Clonogenic Assay

The methylcellulose colony formation assay was used to assess clonogenic potential as previously described [34]. MiaPaCa-2 and PANC-1 pancreatic cancer cells were treated with ASA, OP, and GEM, or their combinations at the indicated concentrations in medium supplemented with 10% FBS for 7 days every 48 h. To assess the clonogenic potential of treated cells at the end of the seven days, cells were trypsinized and resuspended (3 × 10^4^ cells/mL) in 40% methylcellulose supplemented with 1 × DMEM, 1% FBS, and 1% penicillin/streptomycin and plated in 35-mm tissue culture dishes and incubated in 5% CO_2_ at 37 °C. After two weeks, the number of colonies was counted on a phase-contrast microscope. Clonogenicity was determined as the average number of colonies per dish for each treatment group.

### 2.10. Statistical Analysis

Data are presented as the mean ± the standard error of the mean (SEM) from at least three repeats for each experiment performed in triplicate as previously reported by us [35]. Comparisons between two groups from three independent experiments were made by one-way analysis of variance (ANOVA) at 95% confidence using the uncorrected Fisher’s LSD multiple comparisons test with 95% confidence with asterisks denoting statistical significance.

## 3. Results

### 3.1. Aspirin, Oseltamivir Phosphate, and Gemcitabine Preferentially Reduce Pancreatic Cancer Cell Viability in a Concentration-Dependent Manner

ASA exerts its therapeutic effect in both cyclooxygenase (COX)-dependent and -independent pathways to reduce tumor growth and disable tumorigenesis. COX-2 may regulate pancreatic carcinogenesis, with 47–66% of human pancreatic tumors overexpressing COX-2 relative to normal pancreatic tissue [36,37]. COX-2 levels in cancer cells have been reported to be elevated after treatment with 70 nM gemcitabine for 24 h using RT-PCR and Western blot analyses [38]. It is noteworthy that ASA can sensitize cancer cells resistant to GEM, thereby enhancing its therapeutic efficacy [21]. We have also reported that OP treatment of pancreatic cancer targets several important cancer growth factor receptor signaling platforms, oncogenic pathways, and macrophage-mediated tumor progressions with promising therapeutic intent [23].

To determine whether ASA, OP, and GEM reduce pancreatic cancer cell viability without affecting non-malignant cells, the AlamarBlue cytotoxicity assay was used (Figure 1). MiaPaCa-2 and PANC-1 pancreatic cancer cells and HEK 293 cells were treated with ASA, OP, and GEM at increasing concentrations for 72 h to determine the half-maximal inhibitory concentration (IC_50_) values to measure the drug’s efficacy. HEK 293 cells are immortalized non-malignant cells derived from human embryonic kidney cells generated by the transfection of cultures in normal human embryonic kidney cells of either fibroblastic, endothelial, or epithelial cells with sheared adenovirus 5 DNA, and were used as a control to ensure that the proposed treatment was not cytotoxic to non-malignant cells.

For ASA treatment, MiaPaCa-2 cells had an IC_50_ of 0.729 mM, PANC-1 had an IC_50_ of 4.024 mM, and HEK 293 cells had an IC_50_ of 4.753 mM (Figure 1A). Similar trends were observed with OP treatment, where the IC_50_ values were 3.503 mM, 4.696 mM, and 5.026 mM for MiaPaCa-2, PANC-1, and HEK 293 cells, respectively (Figure 1B). As expected, P3ANC-1 cells were more resistant to GEM treatment than MiaPaCa-2 cells, given their more aggressive and naturally resistant nature, with an IC_50_ value of 0.3387 µM compared to 0.286 µM of MiaPaCa-2 (Figure 1C). These values are still lower than the HEK 293 IC_50_ of 0.3854 µM. Consistently, MiaPaCa-2 cells had a lower IC_50_ value, followed by PANC-1 cells and then HEK 293 cells, suggesting that the treatment with ASA, OP, and GEM would not be cytotoxic and preferentially targets malignant cells. 

The combination index (CI) was used to determine the degree of ASA and OP interactions on PANC-1, MiaPaCa-2, and HEK 293 cells. Here, we used the formula of the sum of the ratio of the dose of each drug in the combination to the dose when used alone when the combination and single compound produced 50% (Figure 1 blue bar), 75% (Figure 1 red bar) and 95% (Figure 1 green bar) inhibitory efficacies. CI < 1 indicates synergy, CI = 1 indicates an additive effect, and CI > 1 indicates antagonism. Figure 1D shows the CI values for ASA at concentrations of 0.15, 0.30, and 0.60 mM with increasing OP concentrations of 1.2, 1.6, and 4.8 mM for MiaPaCa-2. Figure 1E shows the CI values for OP at concentrations of 1.2, 1.6, and 4.8 mM with increasing ASA concentrations of 0.15, 0.30, and 0.60 mM for MiaPaCa-2. Figure 1F shows the CI values for GEM at concentrations of 0.0002, 0.0004, and 0.0008 mM with increasing OP concentrations of 1.2, 1.6, and 4.8 mM for MiaPaCa-2. Figure 1G shows the CI values for GEM at concentrations of 0.0002, 0.0004, and 0.0008 mM with increasing ASA concentrations of 0.8, 1.6, and 3.2 mM for MiaPaCa-2. Similarly, Figure 1H,K,L,O show the CI values for ASA, OP and GEM and their combination at indicated concentrations for PANC-1 and HEK 293 cells, respectively. 

It is noteworthy that the combination efficacies of ASA, OP, and GEM are concentration-dependent and have sensitivity differences for each of the cell lines. These findings suggest that the synergistic drug combinations are dependent on their specific concentrations and the characteristic cancer cell type to enable their enhanced therapeutic efficacies. The striking differences in the drug combination indices between PANC-1 and MiaPaCa-2 pancreatic cancer cells may be due to their expression of COX-1 and -2 values through which ASA exerts its therapeutic effect. Omura et al. [39] found that COX-1 and -2 expressions are absent in MiaPaCa-2 cells and in many other pancreatic cancer cells, while PANC-1 cells highly express COX-1 with little expression of COX-2. 

We then investigated the cell survival of the combination treatment of ASA/OP/GEM at or below the identified IC_50_ values for double combinations (ASA+OP, ASA+GEM, and OP+GEM) and triple combination (ASA+OP+GEM). For MiaPaCa-2 cells, combination treatment resulted in a >50% reduction in cell viability in concentrations lower than the individual IC_50_ values (Figure 2A–D). Similarly, in PANC-1 cells, the combination treatment resulted in a >50% reduction in cell viability in concentrations lower than the individual IC_50_ values (Figure 2E–H). For the remainder of the study, MiaPaCa-2 cells were treated with 0.7 mM ASA, 3.5 mM OP, and 0.285 µM GEM, and PANC-1 cells were treated with 4 mM ASA, 4.7 mM OP, and 0.338 µM GEM, which are all lower than the respective IC_50_ values for the control HEK 293 cells. 

### 3.2. Aspirin, Oseltamivir Phosphate, and Gemcitabine Reduce the Metabolic Activity of Pancreatic Cancer Cells

Cancer metabolic activity is significantly related to chemoresistance [40,41]. The specific metabolic alterations in cancer development may be a metabolic functional driver of tumor growth and progression, and as a result, dysregulated metabolic pathways have become attractive targets for cancer therapeutics [42]. Here, we investigated the metabolic activity of MiaPaCa-2, PANC-1, and HEK 293 cells following treatment with ASA, OP and GEM at predetermined IC_50_ values from Figure 1. The AlamarBlue cell viability reagent is an indigo-colored, non-toxic resazurin-based solution that acts as an indicator of cell health by using the reducing power of living cells to quantitatively measure viability. Upon entering living cells, the resazurin is reduced to resorufin, a red compound that is highly fluorescent and provides accurate time-course measurements of metabolic activity of healthy cells with high sensitivity and linearity and involves no cell lysis [43]. The data in Figure 3 reveal that ASA and OP had a similar inhibitory effect on pancreatic cancer cells’ metabolic activity but less so on the HEK 293 immortalized cells. GEM treatment had similar metabolic inhibitor effects on all three cell lines. 

### 3.3. The Combination of Aspirin, Oseltamivir Phosphate, and Gemcitabine Inhibits the Clonogenic Potential of Pancreatic Cancer Cell Lines

Here, we used the methylcellulose clonogenic assay to determine whether there are metastatic, resistant pancreatic progenitors to quantify their ability to proliferate and differentiate into colonies in a semi-solid media. The effect of ASA, OP, and GEM, and their combination treatments on the clonogenicity of MiaPaCa-2 and PANC-1 cells was investigated. MiaPaCa-2 and PANC-1 cells were pre-treated for seven days with ASA, OP, and GEM at predetermined IC_50_ concentrations depicted in Figure 1. 

Figure 4 and Figure 5 show the methylcellulose assay and the data quantification of MiaPaCa-2 and PANC-1 cells, respectively. It is noteworthy that for MiaPaCa-2 cells, the average formed colony size did not significantly differ between GEM treatment alone (99.12 µm) and the untreated control (CTRL) group (105.4 µm). In contrast, the colony sizes for the combinations of ASA+GEM (31.48 µm), OP+GEM (24.08 µm), and ASA+OP+GEM (23.28 µm) resulted in a statistically significant reduction in the average colony diameter compared to the GEM-treated group (*p* < 0.0001) (Figure 4A,B). The number of small, medium, and large colonies formed was significantly lower in all of the combination treatment groups than the untreated and GEM treated groups (Figure 4C–E). This result was consistent with the total number of colonies formed, which was significantly decreased following these treatments, from 283 colonies formed in the untreated group to GEM treatment (129 colonies), ASA+GEM (14 colonies), OP+GEM (10 colonies), and ASA+OP+GEM (3 colonies) (*p* < 0.001), with a significant decrease in colony numbers formed in the combination treatments compared to GEM alone (*p* < 0.01) (Figure 4F).

For PANC-1 cells, the average colony size was significantly reduced with GEM-only treatment (46.04 µm), ASA+GEM (35.28 µm), OP+GEM (25.21 µm), and ASA+OP+GEM (25 µm) compared to the untreated CTRL (82.02 µm) (*p* < 0.0001). Furthermore, the combinations of ASA+GEM (35.28 µm), OP+GEM (25.21 µm), and ASA+OP+GEM (25 µm) resulted in a statistically significant reduction in the average colony diameter compared to the GEM-only treated group (*p* < 0.01) (Figure 5A,B). The number of small, medium, and large colonies formed in the combination treatment groups was significantly lower than the untreated CTRL and GEM treated groups (Figure 5C–E). This was consistent with the data of the total number of colonies formed, which was significantly decreased following treatment, from 308.3 colonies formed in the CTRL group to GEM treatment (40), ASA+GEM (25), OP+GEM (8.667), and ASA+OP+GEM (4.333) (*p* < 0.01), with a statistically significant decrease in the number of colonies formed in the combination treatments compared to GEM alone (*p* < 0.01) (Figure 5F). 

Despite PANC-1 cells being a more aggressive cell line than MiaPaCa-2 cells, it is noteworthy that the untreated and GEM-treated PANC-1 cells had fewer colonies formed compared to their MiaPaCa-2 counterparts. However, it is important to note that the cell lines were treated with different concentrations of ASA/OP/GEM based on the IC_50_ values determined. Nevertheless, as expected, PANC-1 cells demonstrated a smaller clonogenic potential after treatment than their MiaPaCa-2 counterparts, likely due to their increased drug resistance and aggressive nature.

### 3.4. Aspirin, Oseltamivir Phosphate, and Gemcitabine, and Their Combination Modify the Expression of Critical Extracellular Matrix (ECM) Proteins of Pancreatic Cancer Cells

The ECM plays a crucial role in tumorigenesis. Several vital proteins make a significant contribution to the properties of malignant cells and present as possible therapeutic targets. For example, collagen contributes to tumorigenesis, invasion, proliferation, metastasis, the resistance of cancer cell death, and the regulation of intratumoral vessels and hypoxic conditions [44]. Fibronectin is also known to contribute to the hallmarks of cancer, including sustaining proliferation, inducing angiogenesis, inducing invasion and metastasis, avoiding immune destruction, and modulation of cellular energetics [45,46,47]. Interestingly, these ECM proteins, which ultimately promote tumor progression, are disrupted by Neu-1 signaling [47]. Here, we investigated whether treatment with ASA/OP/GEM would impact the expression of some of these essential ECM proteins for MiaPaCa-2 (Figure 6) and PANC-1 cells (Figure 7). 

For collagen, the combination of ASA+OP, OP+GEM, and ASA+OP+GEM significantly decreased the expression of collagen I, collagen II, and collagen IV compared to the untreated CTRL and the GEM-only treated MiaPaCa-2 cells (Figure 6A–C). The combination of ASA+OP, OP+GEM, and ASA+OP+GEM also had the most significant reduction in the expression of fibronectin, laminin, tenascin, and vitronectin compared to the untreated control and the GEM-only treated MiaPaCa-2 cells (Figure 6D–G). For these markers, the combination of ASA+GEM consistently increased the ECM expression markers compared to the untreated controls, suggesting that OP is a critical player in reducing these ECM proteins.

For PANC-1 cells, the combinations of ASA+OP, ASA+GEM, OP+GEM, and ASA+OP+GEM all significantly reduced the expression of all seven ECM proteins compared to both the untreated CTRL and the GEM-only treated cells (Figure 7A–G). The difference in results is primarily the effect of the ASA+GEM combination, which increased expression in MiaPaCa-2 cells but decreased expression in PANC-1 cells and warrants further investigation.

### 3.5. The Combination of Aspirin, Oseltamivir Phosphate, and Gemcitabine Inhibits the Migration of Pancreatic Cancer Cells

Since we showed that treatment with ASA, OP, GEM, and their combination reduces the expression of crucial ECM proteins involved in invasion, migration, and metastasis, we further investigated whether this translated to a reduced migratory capacity of these cells. Using the Fluorometric Chemotaxis Migration Assay Kit, we measured whether a single agent or combination treatment inhibits the migratory capacity of MiaPaCa-2 and PANC-1 pancreatic cancer cells. As shown in Figure 8, ASA and OP alone or in combination with GEM resulted in a significantly lower migratory capacity of both MiaPaCa-2 and PANC-1 cells compared to the untreated CTRL and the GEM-only treated cells. For MiaPaCa-2 cells, the migration, quantified as relative fluorescence units (RFU), went from 7.952 for the CTRL to 3.067 for the ASA+OP+GEM combination, a 61.43% reduction (*p* < 0.0001) (Figure 8A).

For PANC-1 cells, ASA+OP+GEM treatment resulted in a 57.65% reduction in migration potential compared to the CTRL cells and a 39.1% reduction in migration capacity compared to the GEM-only treated cells (*p* < 0.0001) (Figure 8B).

For additional evidence that ASA, OP, and GEM reduce migration of pancreatic cancer cells, the scratch wound assay was performed to measure cell migration in vitro (Figure 9 and Figure 10). Relative wound density (RWD), which measures the density of the wound region relative to the density of the cell region, was used to quantify the migration of pancreatic cancer cells. For MiaPaCa-2 cells, OP, ASA+OP, and ASA+OP+GEM resulted in a significantly reduced RWD compared to the CTRL and GEM-only treated cells at the 24-, 48-, and 72-h timepoints (Figure 9). After 24 h, OP (19.2%), OP+GEM (5.3%), and ASA+OP+GEM (4.0%) had a 74.3, 88.2, and 89.4% lower RWD compared to the CTRL (93.4%), respectively (Figure 9B). The same trend was observed at 48- and 72-h, with OP, ASA+OP, OP+GEM, and ASA+OP+GEM, having a 55.4, 46.9, 52.5, and 63.5% lower RWD compared to CTRL after 48 h, respectively, and a 42.3, 41.2, 35.3, and 53.5% lower RWD compared to CTRL after 72 h (Figure 9C,D).

For PANC-1 cells, OP, ASA+OP, and ASA+OP+GEM were the most effective treatments keeping RWD down compared to CTRL (Figure 10A). After 24 h, OP, ASA+OP, and ASA+OP+GEM had a 62.2, 54.1, and 61.4% reduction in RWD compared to the CTRL (Figure 10B). This same trend was observed at 48- and 72 h as well. OP, ASA+OP, and ASA+OP+GEM had a 73,7, 72.1, and 74.9% reduction in RWD compared to CTRL after 48 h, respectively, and a 77.9, 81.9, and 82.4% reduction in RWD compared to CTRL after 72 h, respectively (Figure 10C,D). 

### 3.6. The Combination of Aspirin, Oseltamivir Phosphate, and Gemcitabine Promotes Apoptosis of Pancreatic Cancer Cell Lines

To assess the effects of the combination of ASA and OP with GEM in inducing apoptotic activity in MiaPaCa-2 and PANC-1 cells, the cells were treated with individual drugs combined with GEM or left as an untreated CTRL for seven days (Figure 11). The early apoptotic cells staining with Annexin V-FITC, apoptotic cells staining with Annexin V-FITC and propidium iodide (PI), and necrotic cells staining with PI following drug treatments were assessed using flow cytometry analysis of the Annexin-V Apoptosis Detection Assay Kit. After treatment, an early time point was used to ensure that a viable number of cells could be analyzed. The assay was used to determine that the treatment of MiaPaCa-2 and PANC-1 cells resulted in increased apoptosis. The premise of the assay is that after drug-initiated apoptosis, phosphatidylserine (PS) from the inner face of the plasma membrane is translocated to the cell surface membrane, which is detected using a fluorescent conjugate of Annexin V [48]. This compound has a high affinity for PS. When stained with PI and Annexin V-FITC, the kit can differentiate between apoptosis and necrosis. 

Mukubou et al. [49] have shown that the treatment of pancreatic cancer cell cultures in vitro and in vivo with GEM and ionizing radiation resulted in synergistic cytotoxicity. After treatment with GEM, the autophagy-related protein light chain 3-II (LC3-II) was upregulated. When GEM was combined with ionizing radiation treatment, LC3-II upregulation was enhanced. Here, MiaPaCa-2 and PANC-1 cells treated with ASA/OP/GEM displayed a significantly greater percentage of apoptotic and necrotic cells compared to the untreated CTRL (Figure 11). As shown in Figure 11A, for MiaPaCa-2 cells, there was a statistically significant increase in the percentage of apoptotic cells in the ASA+GEM (29.65%) cells compared to the CTRL (20.7%), as well as an increased percentage of necrotic cells in the ASA+GEM (18.525%), OP+GEM (19.525%), and ASA+OP+GEM (20%) cells compared to the CTRL (5.4%) and GEM-only (6.67%) cells. In contrast, for PANC-1, there was a statistically significant increase in the percentage of apoptotic cells in the OP+GEM (37.1%) and ASA+OP+GEM (32.9%) cells compared to the CTRL (22.9%) cells (Figure 11B). OP+GEM (26.1%) and ASA+OP+GEM (29.7%) treatment of cells also resulted in a significantly greater percentage of necrotic cells compared to the CTRL (9.5%) and the GEM-only (11.6%) cells. The data in Figure 11 suggest that the treatment of both cell lines with GEM and OP, ASA, and OP+ASA resulted in a significant increased necrosis, likely due to autophagy.

## 4. Discussion

We have previously reported that repurposing ASA as an anti-cancer agent can upend critical targets involved in multistage tumorigenesis regulated by mammalian Neu-1 [22]. Our group has also described a novel signaling paradigm implicated in multistage tumorigenesis [25]. This signaling paradigm involves mammalian Neu-1, which exists in a trimeric complex with MMP-9 and GPCRs. This trimeric complex plays critical roles in ligand-induced activation of several RTKs, including the EGFR [23], insulin receptor (IR) [50], the nerve growth factor (NGF) TrkA receptor [51], and TOLL-like receptors (TLRs) [24,26,52], all of which are upregulated in cancer. We have reported that therapeutic targeting of Neu-1 with OP [28] and ASA [22] disables this intrinsic receptor signaling platform for cancer cell survival in human pancreatic cancer with acquired chemoresistance. Here, we reported on the therapeutic efficacy of ASA and OP in sensitizing and potentiating the efficacy of standard of care GEM for the treatment of pancreatic cancer. The combination of ASA+OP+GEM was found to be most effective at reducing cell viability, clonogenic potential, expression of critical extracellular matrix proteins, migration, and promoting apoptosis. In addition, the synergistic drug combinations are dependent on their specific concentrations and the characteristic cancer cell type to enable their enhanced therapeutic efficacies. The striking differences in the drug combination indices between PANC-1 and MiaPaCa-2 pancreatic cancer cells may be due to their expression of COX-1 and -2 values through which ASA exerts its therapeutic effect. Omura et al. [39] found that COX-1 and -2 expressions are absent in MiaPaCa-2 cells and in many other pancreatic cancer cells, while PANC-1 cells highly express COX-1 with little expression of COX-2. 

There are several reports on ASA and other non-steroidal anti-inflammatory drugs (NSAIDs) combined with chemotherapy [53,54]; however, there was no clear mechanistic explanation for the treatment success. Here, we aimed to identify how the combination of ASA and OP affects some of the critical hallmarks of cancer as outlined by Hanahan and Weinberg; namely, sustain proliferative signaling, growth suppressors evasion, activating invasion and metastasis, enabling replicative immortality, inducing angiogenesis, and resisting cell death [14,15]. ASA, OP, and GEM may also target the emerging hallmarks, including dysregulating cellular genetics and avoiding immune destruction, enabling characteristics of genome instability and mutation, and tumor-promoting inflammation [14]. A review by Zhang et al. [20] highlights the full potential of repurposing non-oncology drugs for clinical cancer management and classifies these candidate drugs into their proposed administration for either mono- or drug combination therapy. Cancer treatment can benefit from anti-inflammatory agents, particularly immunologically cold tumors, although the underlying mechanism(s) remains unclear [55,56]. 

Understanding the key characteristics of the TME and how they relate to the invasive and metastatic TME are also important [57]. Pancreatic cancer is associated with high heterogeneity, metabolic reprogramming, and a dense stromal environment, which result in a high metastatic propensity [58,59,60,61]. Reciprocal communication networks between malignant cells and stromal cells induce changes in the cellular components of the pancreatic TME that prime the primary tumor for metastasis and cell migration [58]. This is the most significant challenge patients face, as most are diagnosed at an advanced disease state with minimal treatment options. Despite GEM being a reference first-line therapeutic option for pancreatic cancer patients since 1997, the five-year survival rate has not improved [62]. It is noteworthy that metastasis can occur even during the early stages of the disease through a stepwise accumulation of genetic and epigenetic alterations [63].

Resistance to chemotherapy treatment is one of the major clinical challenges faced in medical oncology today. One of the more fruitful areas of recent investigation has been the identification of a subpopulation of CSCs within tumors, including pancreatic cancer, which appear to be particularly drug resistant. Studies have documented that cancer cells that undergo EMT can revert to a more drug-resistant CSC phenotype. Cancer treatments have also been shown to foster EMT in surviving cancer cells and promote stem cell enrichment [64]. Considering that drug resistance remains a hallmark of CSCs or cancer cells that have undergone EMT provides a conceptual framework to explore this question further. If GEM resistance is primarily due to the acquisition or pre-existence of an EMT or stem cell phenotype, then reversing the phenotype to a more epithelial, differentiated cancer cell may sensitize cancer cells to treatment with chemotherapy.

This mechanism is particularly relevant to the ASA/OP/GEM treatment we are proposing. We believe that the combination of ASA+OP can interfere with EMT that is triggered by chemotherapy, and this is likely the reason for its effectiveness. In support of this premise, we have previously reported that GEM-resistant PANC-1 (PANC-1-GemR) cells treated with increasing concentrations of ASA (0.1 to 10 mM) revealed a significant concentration- and time-dependent decrease in cell viability [22]. It is noteworthy that resistance to GEM can involve multiple mechanisms, such as altered apoptotic regulating genes and altered expressions or sensitivities of enzyme targets. Many pancreatic cancer cells have no COX-1 expression, with some also lacking COX-2. Rathos et al. [38] reported that COX-2 levels in cancer cells are elevated after treatment with 70 nM GEM for 24 h using RT-PCR and Western blot analyses. Importantly, ASA was reported to sensitize cancer cells resistant to GEM, thereby enhancing the therapeutic efficacy of GEM [21]. 

Interestingly, Guo et al. [65] examined the chemoprotective effects of ASA in a colitis-associated colon cancer model, focusing on the epigenetic histone 3 lysine 27 acetylation (H3K27ac) modulation. The combination of azoxymethane (AOM) and dextran sulfate sodium (DSS) with ASA inhibited AOM/DSS-induced enrichment of H3K27ac in the promoters of inducible nitric oxide synthase (iNOS), tumor necrosis factor-alpha (TNF-α), and interleukin 6 (IL-6) that corresponded to the dramatic suppression of the messenger RNA and protein levels. Furthermore, no significant changes were found in the H3K27ac abundance with the COX-2 promoters or the COX-2 mRNA and protein expression with ASA treatment. Other studies have reported that ASA protects against promoter DNA methylation with an association of the reduced prevalence of E-cadherin (CDH1) promoter methylation in the human gastric mucosa [66]. A total of 33 cellular proteins (including histones) have been identified as targets of ASA-mediated acetylation in colon cancer HCT-116 cells, implying that histone acetylation plays a significant protective role of ASA in colon cancer [67]. The epigenetic modulation by OP in cancer has not been reported.

Advancements in research have revealed that the signaling pathways regulating pancreatic cancer tumorigenesis, including RAS, PI3K/Akt, NFkB, JAK/STAT, Hippo/YAP, and Wnt, have been linked to cancer-related cellular processes of cell proliferation, differentiation, apoptosis, migration, angiogenesis, metabolism, and immune regulation [68]. In line with the previously reported therapeutic effects of ASA, including modulating inflammation and critical signaling pathways and proteins such as Wnt/β- catenin signaling, AMP-activated protein kinase (AMPK), mammalian target of rapamycin (mTOR), p53, NF-kB signaling, and Bcl-2 [21,69,70], the findings in the present study support ASA as a promising anticancer repurposed drug. 

Furthermore, hypovascularization and a dense desmoplasia create a highly hypoxic and nutrient-limited microenvironment [58]. Endothelial cells represent a physical connection between the circulatory system and tumor cells, with endothelial cell adhesion proteins being essential for immune cell recruitment and are frequently downregulated in tumor-associated vasculature. Interestingly, immune-cold tumors had reduced expression of endothelial adhesion proteins but showed elevated activity in VEGF and hypoxia pathways, which are integral to the remodeling of endothelial cells during tumorigenesis. Collectively, endothelial cell remodeling, accompanied by elevated VEGF and hypoxia pathways, increased glycolysis, and cell junction dysregulation might collectively inhibit immune cell infiltration and function [71].

Collagen is the major TME component, which increases tumor tissue stiffness and modulates tumor immunity to promote metastasis. There is a feedforward loop with collagen and cancer cells, with collagen influencing cancer cell behavior to exacerbate cancer progression, which reshapes the collagen to promote cancer progression further. Collagen-rich ECM results in the binding and recruitment of other molecules to form dense fibrosis, influencing the oxygen level within the TME and modifying the integrity of the new vasculature [44]. Here, we showed that treatment with ASA+OP+GEM significantly decreased expression of collagen I and decreased collagen II and IV expression in MiaPaCa-2 cells and significantly decreased expression of collagen I, II, and IV in PANC-1 cells (Figure 6 and Figure 7, respectively). 

Fibronectin has a controversial role in cancer progression, with reports of having both a tumor-suppressive role as well as a pro-metastatic role. Hypoxia is often considered at the crossroads between early progression and late malignancy, relating to fibronectin’s different roles [46]. As tumors grow, the TME becomes more hypoxic, resulting in an environmental pressure that promotes cell death and selects a small number of cells with more stem cell-like properties, including fibronectin expression, to survive. Concurrently, tumor cells undergoing EMT develop more mesenchymal phenotypes. The cancer-associated fibroblasts (CAFs) become fibrinolytic, resulting in fibronectin’s clearance in the ECM, making space for the growing tumor and opening the roadmap for cancer metastasis [72]. The present study demonstrated that ASA+OP+GEM treatment significantly reduced fibronectin expression in MiaPaCa-2 and PANC-1 cells (Figure 6 and Figure 7). This may be due to the reduced clonogenic potential of cells following treatment, suggesting that ASA+OP+GEM treatment preferentially targets malignant cells with stem cell-like properties, such as regenerative capacity (Figure 4 and Figure 5). Given that we have reported on the role of OP [23,28] and ASA [22] in targeting and shutting down Neu-1 activity in complex with growth factor receptors, these findings are not surprising. A recent study reported on Neu-1 suppressing bladder cancer progression by inhibiting the fibronectin-integrin α5β1 interaction and the Akt signaling pathway [47]. Neu-1 overexpression was reported to decrease cell viability and increase apoptosis in bladder cancer cell lines. Furthermore, given the role of α5β1 in protecting against apoptosis and promoting growth through the PI3K/Akt pathway, Neu-1 was found to promote cell proliferation and induce apoptosis by inhibiting the Akt pathway, which warrants further investigation for pancreatic cancer cell lines.

On the other hand, vitronectin is the key controller of mammalian tissue repair and remodeling activity, playing critical roles in thrombogenesis and tissue repair [73]. By interacting with cell-surface receptors, vitronectin triggers signaling cascades that affect cell attachment, migration, and survival, playing a pivotal role in tissue repair. This is particularly relevant for cancer, as it is often referred to as a wound that does not heal. Figure 6 and Figure 7 depict a reduction in vitronectin expression following treatment with ASA+OP+GEM in both pancreatic cancer cell lines, suggesting that combination treatment disrupts the attachment and migration capacity of malignant cells. These results are further supported by the migration and scratch wound assays (Figure 8, Figure 9 and Figure 10). The migration capacity of both MiaPaCa-2 and PANC-1 cells was significantly reduced in cells treated with ASA+OP+GEM compared to the control and GEM-only treated cells (Figure 8). This was mirrored in the scratch wound assay. The relative wound density of MiaPaCa-2 and PANC-1 cells treated with ASA+OP+GEM was significantly lower than the control, and GEM-only treated cells (Figure 9 and Figure 10, respectively).

Laminins are large extracellular glycoprotein components of basement membranes. They are involved in several biological processes, including cellular interactions, self-polymerization, and binding with other extracellular matrices’ (ECM) proteins [74]. Tenascins are large oligomeric glycoproteins of the ECM, synthesized at specific times and locations during embryonic development, have restricted locations in adult tissues [75], and are prominently expressed in solid tumors [76]. Figure 6 and Figure 7 depict a significant reduction in laminin and tenascin expressions following treatment with ASA+OP+GEM in both pancreatic cancer cell lines.

Over several decades, altered sialylation of tumor cell surface glycoproteins has been highly associated with the cancer progression and metastatic phenotype [77,78]. Tumor-derived sialic acids have been shown to disable cytotoxicity mechanisms of effector immune cells, trigger the production of immune-suppressive cytokines and dampen the activation of antigen-presenting cells, with aberrant sialylation favoring tumor growth and progression [25,79]. Since glycosylation is related to each of the hallmarks of cancer, it can be considered a hallmark of cancer [80]. More specifically, *N*-linked glycoproteins are upregulated in tumors. Biosynthesis of *N*-linked glycoproteins is regulated by the glycoprotein substrates and glycosylation enzymes for glycan synthesis and conjugation to glycoproteins. *N*-linked glycoproteins upregulation was modified by complex glycans with sialic acids or fructose. Focusing on sialylated and fucosylated glycans of the *N*-linked glycoproteins upregulated in PDAC can increase the specificity of the markers for cancer [71]. We reported aspirin [22] and OP [28] target and inhibit Neu-1 activity. As with all neuraminidases, Neu-1 hydrolyzes links on growth factor receptors for cancer cells, making Neu-1 a vital regulator of glycosylated receptors [81]. Altered glycosylation of growth factor receptors affects cancer cell signal transduction pathways, including modulation of tumor cell growth and proliferation. Investigating and targeting the glycosylation status of these receptors may resolve the challenges faced by current targeted therapy options.

## 5. Conclusions

Drug repurposing is a trend that opens a discovery window for including them in cancer therapy and can vastly improve our ability to treat cancer more effectively. A quote from James Black, Nobel Laureate in physiology and medicine, states that “the most fruitful basis for discovering a new drug is to start with an old drug” [82]. This report presents a novel therapeutic effect of the combination of aspirin and oseltamivir phosphate with chemotherapeutic gemcitabine as a treatment option for pancreatic cancer. These data suggest that targeting Neu-1 on pancreatic cancer cells with these repurposed drugs is crucial for modulating cell proliferation, combination efficacies, invasion, clonogenicity, and migration (graphical abstract). These promising results warrant additional investigation to assess the potential of translating into the clinical setting to improve the cancer patient prognosis for an otherwise fatal disease. 

## Figures and Tables

**Figure 1 cancers-14-01374-f001:**
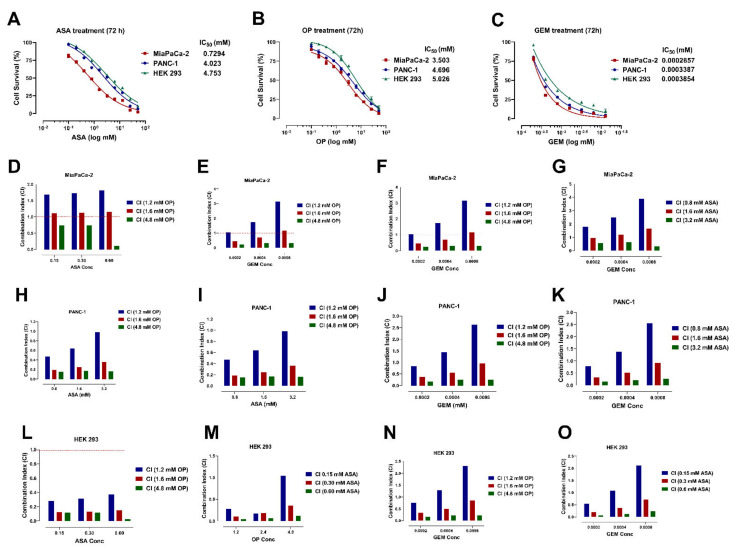
ASA, OP, GEM, and their combination reduce pancreatic cancer cell viability concentration-dependently. MiaPaCa-2, PANC-1, and HEK 293 cells, as immortalized non-malignant control cells, were plated in 96-well plates at 20,000 cells/well and treated with increasing concentrations of (**A**) ASA, (**B**) OP, and (**C**) GEM for 72 h to identify individual IC_50_ values. The combination index (CI) was calculated to determine the degree of ASA and OP interactions, and their interactions with GEM on MiaPaCa-2 (**D**–**G**), PANC-1 (**H**–**K**), and HEK 293 (**L**–**O**) cells. The sum of the ratio of the concentration (mM) of each drug in the compound to the dose when used alone when the combination and drugs produce 50% (blue bar), 75% (red bar), and 95% (green bar) efficacies. The formula of the combination index (CI) = (D)1/(Dχ)1 + (D)2/(Dχ)2, where (Dχ)1 and (Dχ)2 represented concentrations (mM) of each drug alone to exert x% efficacies, while (D)1 and (D)2 are concentrations (mM) of the drugs in combination to elicit the same effect. CI < 1, = 1 and > 1 indicate synergism, additivity, and antagonism, respectively. Data are presented as mean ± SEM compared to the untreated control at day 0. Data are represented as the mean ± SEM of 3 independent experiments performed in triplicates. Abbreviations: ASA, aspirin; OP, oseltamivir phosphate; GEM, gemcitabine; IC_50_, half-maximal inhibitory concentration; SEM, standard error of the mean; CI, combination index.

**Figure 2 cancers-14-01374-f002:**
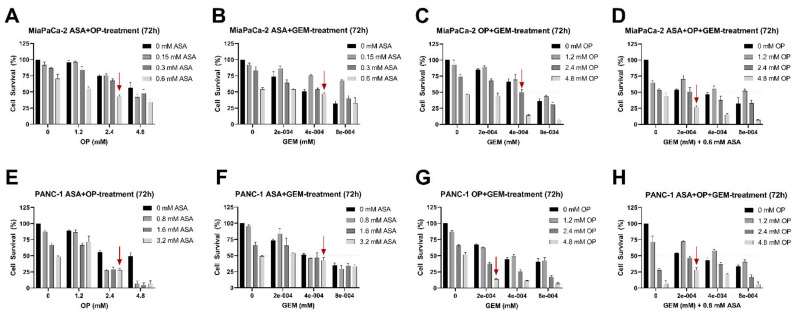
ASA, OP, GEM, and their combination reduce pancreatic cancer cell viability concentration-dependently. MiaPaCa-2 cells were plated in 96-well plates at 20,000 cells/well and treated with combinations of (**A**) ASA+OP, (**B**) ASA+GEM, (**C**) OP+GEM, (**D**) ASA+OP+GEM at increasing concentrations for 72 h. PANC-1 cells were plated in 96-well plates at a density of 20,000 cells/well and treated with increasing concentrations of (**E**) ASA+OP, (**F**) ASA+GEM, (**G**) OP+GEM, and (**H**) ASA+OP+GEM at increasing concentrations for 72 h. The red arrows represent the first combination therapy that achieves cell viability of <50% at or below the single-agent IC_50_ values. Data are presented as mean ± SEM compared to the untreated control at day 0. Data are represented as the mean ± SEM of 3 independent experiments performed in triplicates. Abbreviations: IC_50_, half-maximal inhibitory concentration; ASA, aspirin; OP, oseltamivir phosphate; GEM, gemcitabine; SEM, standard error of the mean.

**Figure 3 cancers-14-01374-f003:**
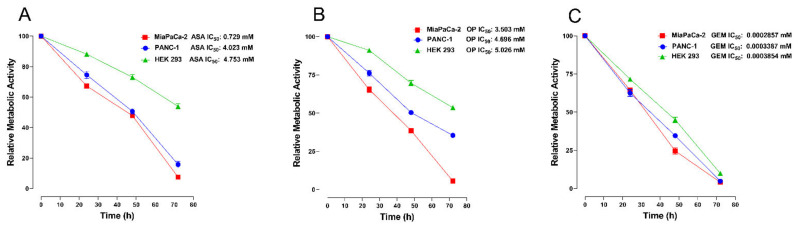
The effect of ASA, OP, and GEM treatment on the metabolic activity of pancreatic cancer cells at IC_50_ values using the AlamarBlue live cell assay. MiaPaCa-2, PANC-1, and HEK 293 cells, as immortalized non-malignant control cells, were plated in 96-well plates at 20,000 cells/well density and treated with indicated IC_50_ concentrations of (**A**) ASA, (**B**) OP, and (**C**) GEM for 72 h to identify metabolically active cells following drug treatment. Data are presented as mean ± SEM of 3 independent experiments performed in triplicates. Abbreviations: IC_50_, half-maximal inhibitory concentration; ASA, aspirin; OP, oseltamivir phosphate; GEM, gemcitabine; SEM, standard error of the mean.

**Figure 4 cancers-14-01374-f004:**
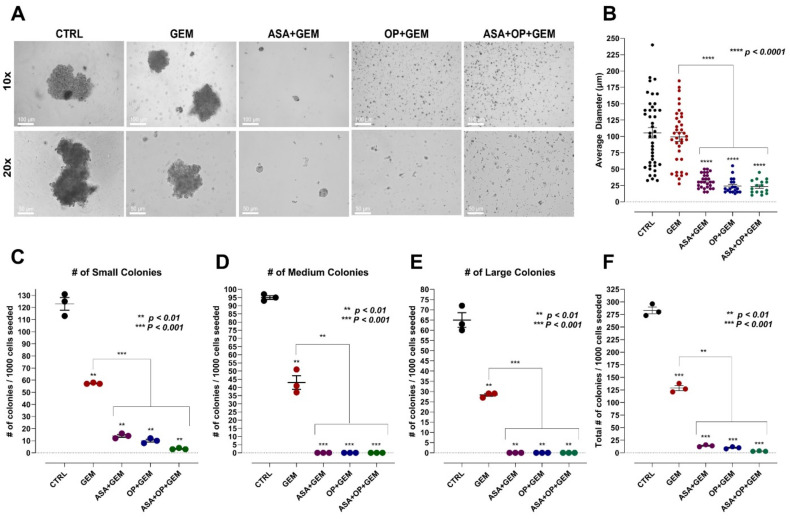
ASA, OP, GEM, and their combination reduce the clonogenic potential of MiaPaCa-2 pancreatic cancer cells. MiaPaCa-2 cells were treated with ASA (0.7 mM), OP (3.5 mM), GEM (0.285 µM), or their combination for 7 days. Cells were counted and resuspended in methylcellulose-containing media at a density of 3000 cells/mL. Cells were plated in 35-mm tissue culture dishes and incubated for 14 days. (**A**) Images were obtained after 14 days using phase microscopy, with the 10× and 20× objective. (**B**) The diameter of colonies was measured using a graded dish on a phase-contrast microscope. The number of (**C**) small colonies (<50 µm), (**D**) medium colonies (>50 µm;<100 µm), and (**E**) large colonies (>100 µm) was plotted. (**F**) The degree of clonogenicity was determined as the total number of colonies per dish in triplicates. The number of colonies formed from the treatments was compared to the CTRL and GEM-treated groups using the one-way ANOVA Fisher test comparisons with 95% confidence, indicating asterisks for statistical significance. Data are presented as the mean ± SEM of 3 independent experiments performed in triplicates. Abbreviations: ASA, aspirin; OP, oseltamivir phosphate; GEM, gemcitabine; CTRL, control; SEM, standard error of the mean.

**Figure 5 cancers-14-01374-f005:**
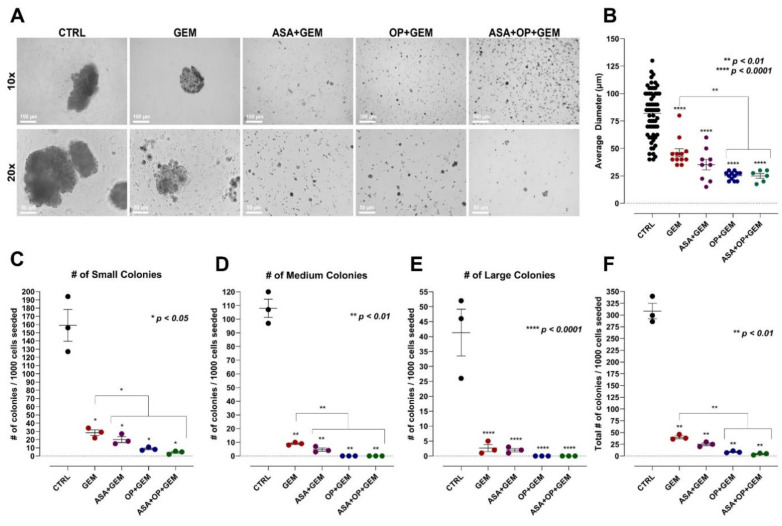
ASA, OP, GEM, and their combination reduce the clonogenic potential of PANC-1 pancreatic cancer cells. PANC-1 cells were treated with ASA (4 mM), OP (4.7 mM), GEM (0.338 µM), or their combination for 7 days. Cells were counted and resuspended in methylcellulose-containing media at a density of 3000 cells/mL. Cells were plated in 35-mm tissue culture dishes and incubated for 14 days. (**A**) Images were obtained after 14 days using phase microscopy, using the 10× and 20× objective. (**B**) The diameter of colonies was measured using a graded dish on a phase-contrast microscope. The number of (**C**) small colonies (<50 µm), (**D**) medium colonies (>50 µm; <100 µm), and (**E**) large colonies (>100 µm) was plotted. (**F**) The degree of clonogenicity was determined as the total number of colonies per dish in triplicates. The number of colonies formed from the treatments was compared to the CTRL and GEM-treated groups using the one-way ANOVA Fisher test comparisons with 95% confidence, indicating asterisks for statistical significance. Data are presented as mean ± SEM of 3 independent experiments performed in triplicates. Abbreviations: ASA, aspirin; OP, oseltamivir phosphate; GEM, gemcitabine; CTRL, control; SEM, standard error of the mean.

**Figure 6 cancers-14-01374-f006:**
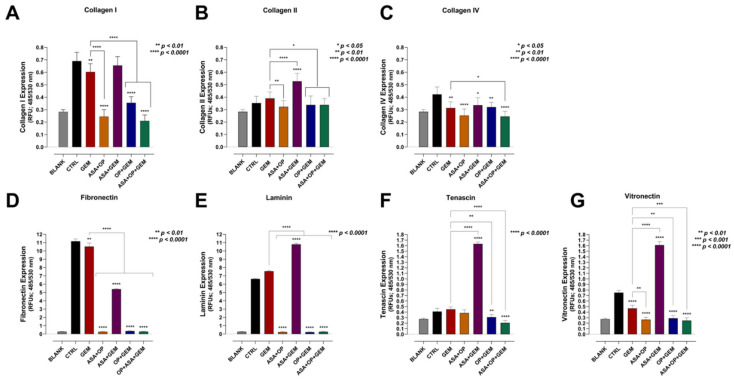
ASA, OP, GEM, and their combination alter the expression of adhesion markers in MiaPaCa-2 cells. MiaPaCa-2 cells were pre-treated with ASA (0.7 mM), OP (3.5 mM), GEM (0.285 µM), or their combination for 7 days. Cells were collected, resuspended, and adhered to coated substrates to capture adherent cells. Adherent cells were lysed and dyed, allowed to incubate, and read with a fluorescence plate reader. Graphs show expression of (**A**) collagen I, (**B**) collagen II, (**C**) collagen IV, (**D**) fibronectin, (**E**) laminin, (**F**) tenascin, and (**G**) vitronectin as relative fluorescent units (RFUs) (ex 450 nm: em 530 nm). Data are presented as the mean ± SEM of 3 independent experiments performed in triplicates. Significance is shown compared to the CTRL unless otherwise indicated using the one-way ANOVA Fisher test comparisons with 95% confidence with indicated asterisks for statistical significance. Abbreviations: ASA, aspirin; OP, oseltamivir phosphate; GEM, gemcitabine; RFU, relative fluorescent unit; SEM, standard error of the mean; CTRL, control.

**Figure 7 cancers-14-01374-f007:**
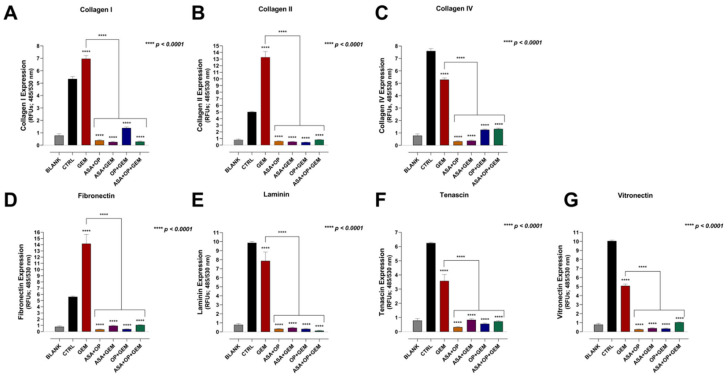
ASA, OP, GEM, and their combination alter the expression of adhesion markers in PANC-1 cells. PANC-1 cells were pre-treated with ASA (4 mM), OP (4.7 mM), GEM (0.338 µM), or their combination for seven days. Cells were collected, resuspended, and adhered to coated substrates to capture adherent cells. Adherent cells were lysed and dyed, allowed to incubate, and read with a fluorescence plate reader. Graphs show expression of (**A**) collagen I, (**B**) collagen II, (**C**) collagen IV, (**D**) fibronectin, (**E**) laminin, (**F**) tenascin, and (**G**) vitronectin as relative fluorescent units (RFUs) (Ex 450 nm; em 530 nm). Data are presented as the mean ± SEM of 3 independent experiments performed in triplicates. Significance is shown compared to the CTRL, unless otherwise indicated, using the one-way ANOVA Fisher test comparisons with 95% confidence with indicated asterisks for statistical significance. Abbreviations: ASA, aspirin; OP, oseltamivir phosphate; GEM, gemcitabine; RFU, relative fluorescent unit; SEM, standard error of the mean; CTRL, control.

**Figure 8 cancers-14-01374-f008:**
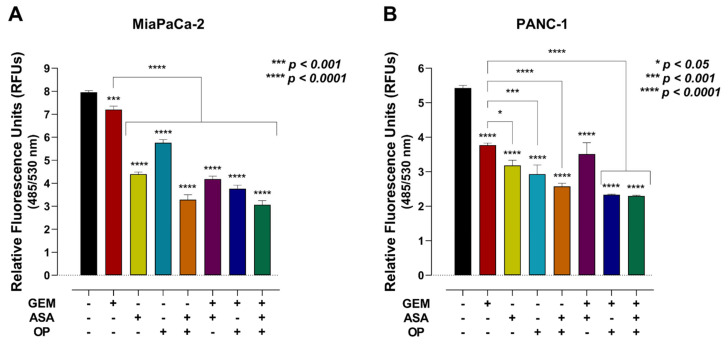
ASA, OP, GEM, and their combination inhibit migration of MiaPaCa-2 and PANC-1 pancreatic cancer cells. (**A**) MiaPaCa-2 cells were pre-treated with ASA (0.7 mM), OP (3.5 mM), GEM (0.285 µM), or their combination, and (**B**) PANC-1 cells were pre-treated with ASA (4 mM), OP (4.7 mM), GEM (0.338 µM), or their combination for 7 days. Treated cells were collected and resuspended in 1× DMEM (no FBS), added into the migration chamber, and allowed to adhere for 4 h to allow for migration into the feeder tray containing 10% FBS. Unbound cells were removed, and the migration insert plate was transferred onto a new cell culture tray with the cell detachment solution. Cells were incubated for 30 min, and the CyQuant GR dye in lysis buffer was added to the wells and incubated for 15 min. The mixture was transferred to a 96-well plate and read with a fluorescence plate reader. The data are presented as relative fluorescence units (RFUs) (ex 485; em 530) ± SEM of 3 independent experiments performed in triplicates. Significance is shown compared to the CTRL unless otherwise shown. Abbreviations: ASA, aspirin; OP, oseltamivir phosphate; GEM, gemcitabine; FBS, fetal bovine serum; RFU, relative fluorescence unit; SEM, standard error of the mean; CTRL, control.

**Figure 9 cancers-14-01374-f009:**
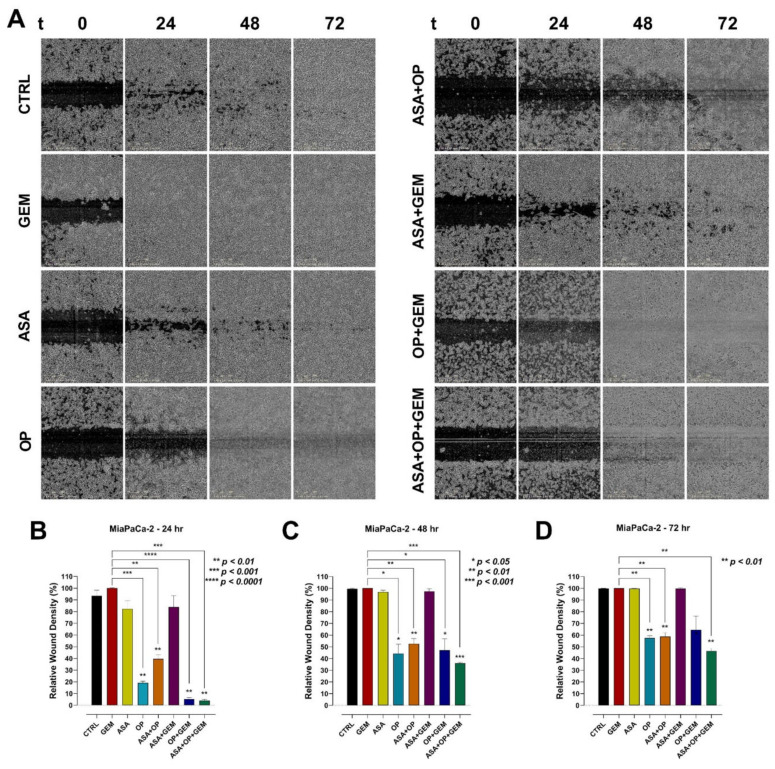
ASA, OP, GEM, and their combination inhibit wound closure of MiaPaCa-2 pancreatic cancer cells. MiaPaCa-2 cells were seeded in a flat bottom ImageLock™ 96-well plate and allowed to adhere overnight. The WoundMaker™ was used to create reproducible wounds in all wells. Unbound cells were washed, and cells were treated with ASA (0.7 mM), OP (3.5 mM), GEM (0.285 µM), or their combination. (**A**) The IncuCyte ZOOM^®^ software was used to scan the plate repeatedly over 72 h. The wound migration area was quantified using the IncuCyte ZOOM^®^ software comparing the wound density at (**B**) 24 h, (**C**) 48 h, and (**D**) 72 h to the immediate wound density. The data are presented as relative wound density ± SEM of 3 independent experiments performed in triplicates. Significance is shown compared to the CTRL unless otherwise shown. Abbreviations: ASA, aspirin; OP, oseltamivir phosphate; GEM, gemcitabine; SEM, standard error of the mean; CTRL, control.

**Figure 10 cancers-14-01374-f010:**
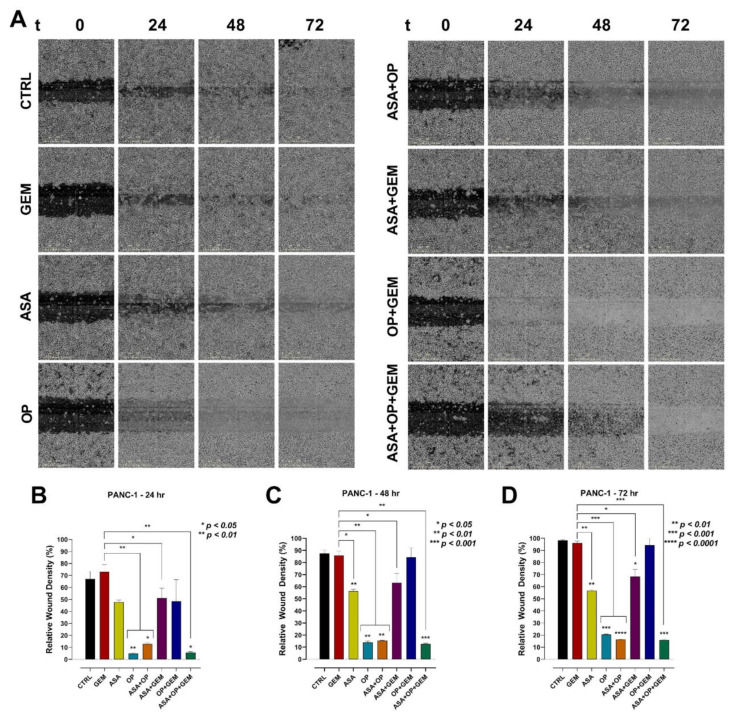
ASA, OP, GEM, and their combination inhibit wound closure of PANC-1 pancreatic cancer cells. PANC-1 cells were seeded in a flat bottom ImageLock™ 96-well plate and allowed to adhere overnight. The WoundMaker™ was used to create reproducible wounds in all wells. Unbound cells were washed, and cells were treated with ASA (4 mM), OP (4.7 mM), GEM (0.338 µM), or their combination. (**A**) The IncuCyte ZOOM^®^ software was used to scan the plate repeatedly over 72 h. The wound migration area was quantified using the IncuCyte ZOOM^®^ software comparing the wound density at (**B**) 24 h, (**C**) 48 h, and (**D**) 72 h to the immediate wound density. The data are presented as relative wound density ± SEM of 3 independent experiments performed in triplicates. Significance is shown compared to the CTRL unless otherwise shown, using the one-way ANOVA Fisher test comparisons with 95% confidence with indicated asterisks for statistical significance. Abbreviations: ASA, aspirin; OP, oseltamivir phosphate; GEM, gemcitabine; SEM, standard error of the mean; CTRL, control.

**Figure 11 cancers-14-01374-f011:**
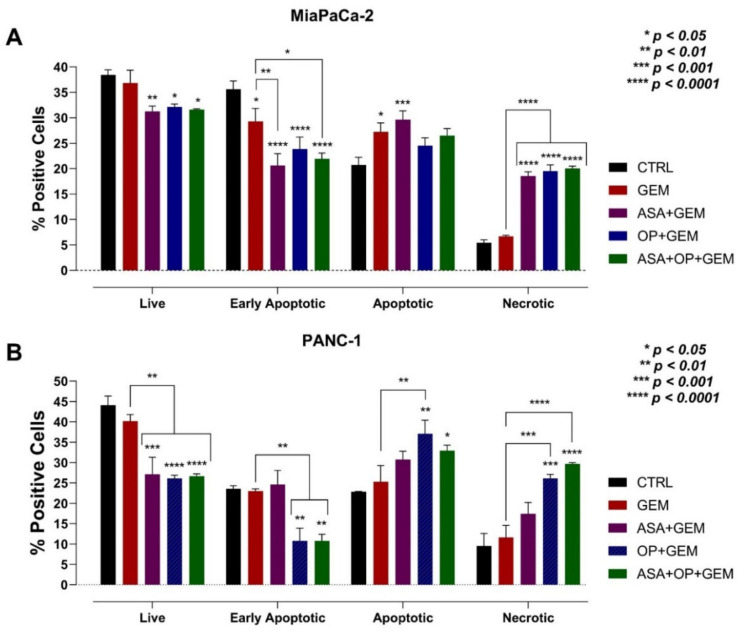
Viability, early apoptosis, apoptosis, and necrosis of MiaPaCa-2 and PANC-1 cells after treatment with ASA, OP, GEM, and their combination using the Annexin V-FITC and Propidium Iodide Assay. (**A**) MiaPaCa-2 cells were pre-treated with ASA (0.7 mM) and OP (3.5 mM) together with GEM (0.285 µM), or all in combination, and (**B**) PANC-1 cells were pre-treated with ASA (4 mM), OP (4.7 mM), together with GEM (0.338 µM), or all in combination for seven days. Treated cells were collected and resuspended in binding buffer and incubated with Annexin V-FITC and propidium iodide (PI) for 5 min at room temperature. Apoptotic and necrotic cells were analyzed by flow cytometry and presented as the mean ± SEM of the percentage of positive cells of 3 independent experiments performed in duplicates, with early apoptotic cells staining for Annexin V-FITC, apoptotic cells staining for Annexin V-FITC and PI, and necrotic cells staining for PI. Significance is represented compared to the CTRL and GEM-only treated cells by one-way ANOVA using the uncorrected Fisher’s LSD multiple comparisons test with 95% confidence indicated asterisks for statistical significance. Abbreviations: ASA, aspirin; OP, oseltamivir phosphate; GEM, gemcitabine; PI, propidium iodide; SEM, standard error of the mean; CTRL, control.

## Data Availability

All data needed to evaluate the conclusions in the paper are present in the paper.

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
