# Peer review of "Next Generation of Cancer Drug Repurposing: Therapeutic Combination of Aspirin and Oseltamivir Phosphate Potentiates Gemcitabine to Disable Key Survival Pathways Critical for Pancreatic Cancer Progression"

_cancers, 2022, doi:10.3390/cancers14061374_

Round 1

Reviewer 1 Report

In general the manuscript is well written and presents the results in a proper way. Overall, the study is well described, aims and results are depicted in a clear way. However, there are some points which are not described well enough but would be necessary for interpretation.

1- Authors should describe why they chose those two pancreatic cell line. Are they representative of the disease entity ? I mean both are K-ras mutated, TP53 mutated, Smad4 WT and CDKN2A deleted.

In section 2.1 and figure 1 Authors showed that PDAC cell lines could be sensitized by the association of ASA/OP/Gem. It would interesting to show if association have additive or synergetic effect compared to each treatment alone. The could be done by evaluating the combination index (CI) by the Chou-Talalay equation, which takes into account both the potency and the shape of the dose-effect curve. Thus I also suggest to present dose response curve in semi-log scale (Figure 1A-B-C). Authors evaluated each treatment in HK 293 cells as well but do not show the association effect. Although concentrations are lower than the IC50 of HK 293, drugs association could show a potential effect on non malignant cell line.

In section 2.2 and figure 2, Authors showed that treatment impairs metabolic activity by using Alamar-Blue. The same assay was used for viability evaluation. how does authors could be sure that drugs decrease viability and not just metabolism ? Indeed, in the section 2.3 and figure 3-4, authors showed that drugs impaired clonogenicity and PANC-1 cells seems to be more sensitive to drugs than Mia Paca-2 cells, which not seems to be consistent with previous data (section 2.1).

In section 2.6, authors describe apoptosis assay. could authors be more explicit about how they differentiate late apoptosis and necrosis. Indeed, early apoptosis are PI negative/annexin positive, and late apoptosis and necrosis are PI+/Annexin+. All cell line are TP53 mutated. is apoptosis the main type of cell death ? It has been shown that autophagy could be induced by gemcitabine. can authors discuss the type of cell death ?

Methods :

cell line : Authors should describe the normal cell line HK 293

The authors should consider to conduct in vivo experiments to verify their findings. Indeed, authors hypothesized that OP should alter metastatic progression. This should be tested and could be a real game changer as metastatic progression is very frequent in PDAC.

Authors detailed a molecular mechanism of sensitization. they could describe how the repurposed drugs could overcome gemcitabine resistance or how it could be synergistic with gemcitabine.

Reviewer 2 Report

  1. For Results 2.1 & 2.2, suggest to use primary normal HPDE cells or immortalized HPDE/E6E7 cells instead of kidney cells to represent the normal pancreatic cells.
  2. Fig. 1 H-K, the ASA+OP+GEM triple combination didn't show the better effect than OP+GEM, please explain the possibility. 
  3. The full name of HEK 293 should be mentioned in Results 2.1 but not 2.2.
  4. In Results 2.2, the authors claimed that ASA, OP, GEM reduced the metabolic activity of pancreatic cancer cells. However, the method the authors used was alamarBlue cell viability reagent, it's just one of the cell viability assays, like MTT assay or MTS assay, can only represent the population of living cells but not the metabolic status of the cells. It's not a proper way to measure the metabolic activity by such method. Besides, the references the authors cited to address "Cancer metabolic activity is significantly related to chemoresistance", are the studies about the reprogramming of energy metabolism and to identify the novel biosynthetic pathways (ref 35), and the review article discussing the desmoplastic reaction, extracellular acidosis and hypoxia in tumor microenvironment in pancreatic cancer (ref 36). If the authors wanted to study the effects of ASA, OP, GEM on metabolic activity, much more conscientious experiments are necessary. 
  5. The first paragraph of Results 2.3 mentioned about the EGFR mutation, however, the role of EGFR in the present work was never studied or addressed. The rationale of Results 2.3 is needed to be clarified.
  6. In Fig. 4B, the colony size of the GEM treatment group seems significantly smaller than the control group with p value < 0.0001. Why the authors said the average colony size did NOT significantly differ between the GEM treatment (46.04 µm) and the untreated CTRL (82.02 µm)?
  7. In Results 2.4, the authors mentioned the combination modify the expression of ECM proteins, however, there're no any invasion-related experiments conducted. Chemotaxis migration assay and scratch wound assay can only measure the migratory ability but not the interplay between cancer cells and ECM. The matrigel invasion assay is necessary to support the findings in Results 2.4.
  8. The animal study is required to prove the combination effects of aspirin and oseltamivir phosphate could potentiate gemcitabine for pancreatic cancer treatment in vivo.

Round 2

Reviewer 2 Report

  1. Although the authors self-cited their previous findings on EGFR regulaition as background, the role of EGFR in 2.3 remains unclarified. I'd suggest just focus on the importance of clonogenic potential in carcinogenesis and mention the inhibitory effect of combined treatment on colony formation. 'cause the mention of EGFR or even Neu-1 in this section is quite confusing.
  2. Some preliminary in vivo data still needs to be provied to make current manuscript be accepted.

Author Response

Comments and Suggestions for Authors

  1. Although the authors self-cited their previous findings on EGFR regulaition as background, the role of EGFR in 2.3 remains unclarified. I'd suggest just focus on the importance of clonogenic potential in carcinogenesis and mention the inhibitory effect of combined treatment on colony formation. 'cause the mention of EGFR or even Neu-1 in this section is quite confusing.

Author’s response:  Thank you for this comment. We have removed the text on EGFR regulation and Neu1 to improve the clarity to avoid confusion. This section starts with "Here, we used the methylcellulose clonogenic assay to determine whether there are metastatic, resistant pancreatic progenitors to quantify their ability to proliferate and differentiate into colonies in a semi-solid media." 

  1. Some preliminary in vivo data still needs to be provied to make current manuscript be accepted.

Author’s response:  Thank you for this important comment. The previous two peer-reviewers commented on in vivo studies and they have accepted our response as follows:  

"We have completed an extensive preclinical animal study of human PANC-1 xenografts and another manuscript is now in preparation for consideration for publication.

The experimental design of the preclinical animal studies was to use ASA and OP in an osmotic pump surgically implanted near the tumor. The pump is to provide a continuous perfusion of the drugs on an hourly basis for an optimal dosage per day for 42 days. The animals also received GEM once a week. The premise of the pump is to block the compensatory reaction of the tumor to the drugs and GEM. We found that the animals with the (ASA+OP) pump had reduced the tumor volume by 85%, prevented GEM-resistance compared to the GEM-only cohort, prevented metastasis to the liver and lungs and had long term survival rate for 100 days with no adverse side effects. Indeed, this innovative procedural treatment for pancreatic cancer could be a potential game changer for patients with advanced pancreatic cancer."

Round 3

Reviewer 2 Report

I have no further comments or suggestions.